# Application of Pyrolysis for the Evaluation of Organic Compounds in Medical Plastic Waste Generated in the City of Cartagena-Colombia Applying TG-GC/MS

**DOI:** 10.3390/ijms24065397

**Published:** 2023-03-11

**Authors:** Joaquín Hernandez-Fernandez, Henry Lambis, Ricardo Vivas Reyes

**Affiliations:** 1Chemistry Program, Department of Natural and Exact Sciences, San Pablo Campus, University of Cartgena, Cartagena 130015, Colombia; 2Chemical Engineering Program, School of Engineering, Universidad Tecnológica de Bolivar, Parque Industrial y Tecnológico Carlos Vélez Pombo Km 1 Vía Turbaco, Cartagena 130001, Colombia; 3Department of Natural and Exact Science, Universidad de la Costa, Barranquilla 080002, Colombia; 4Fundacion Universitaria Tecnologico Comfenalco—Cartagena, Cr 44 D N 30A, 91, Cartagena 130001, Colombia

**Keywords:** polymers, wastes, degradation, analyses, composition

## Abstract

In this study, the thermal degradation and pyrolysis of hospital plastic waste consisting of polyethylene (PE), polystyrene (PS), and polypropylene (PP) were investigated using TG-GC/MS. The identified molecules with the functional groups of alkanes, alkenes, alkynes, alcohols, aromatics, phenols, CO and CO_2_ were found in the gas stream from pyrolysis and oxidation, and are chemical structures with derivatives of aromatic rings. They are mainly related to the degradation of PS hospital waste, and the alkanes and alkenes groups originate mainly from PP and PE-based medical waste. The pyrolysis of this hospital waste did not show the presence of derivatives of polychlorinated dibenzo-p-dioxins and polychlorinated dibenzofurans, which is an advantage over classical incineration methodologies. CO, CO_2_, phenol, acetic acid and benzoic acid concentrations were higher in the gases from the oxidative degradation than in those generated in the pyrolysis with helium. In this article, we propose different pathways of reaction mechanisms that allow us to explain the presence of molecules with other functional groups, such as alkanes, alkenes, carboxylic acids, alcohols, aromatics and permanent gases.

## 1. Introduction

Polymers have helped satisfy human needs and are gaining significant importance on a daily basis due to their multiple applications [1,2,3,4,5]. Polymers are widely used in manufacturing automotive parts, household appliances, electrical instruments, food containers, homes, medical products, water purification and many other applications [1,2,3,4,5,6,7,8]. Their many applications are clearly in great demand, based on the fact that the annual production of polymers ranges from 359 to 380 million tons per year. Of the production volumes, a significant percentage ends up in various bodies of water and it becomes one of the most critical environmental pollutants on the planet [3,4,5,9,10,11]. Currently, one of the most serious types of harm that are caused by plastic is the generation of bioaccumulative microplastics [12,13,14,15]. In the United States alone, hospitals were reported in 2020 to generate 5.9 million tons of medical waste per year [16]. 70% of medical devices are made of some type of polymer, for example: polyethylene (PE), polystyrene (PS), or polypropylene (PP), among others, and it is for this same reason that these materials are one of the main components present in hospital plastic waste (HPW) [17]. HPW is not a clean or pure plastic that can be quickly recovered or processed as it is impregnated with infectious biological residues, chemical traces, pharmaceutical residues, sensitive materials, and cytotoxic and radioactive substances [7,16,18,19,20]. This means that HPW is among the most significant contributors to global waste [3,19] since it is part of the waste that usually results from the routine activity of medical centers, clinics, hospitals, and any institution or organization related to health, specifically patient care [16,18,19]. Therefore, 10% and 25% contain chemical species classified as infectious, toxic, and polluting. Despite being a low percentage, these residues need special handling to ensure their safe disposal [17]. Improper handling makes them a potential source of infection or injury for patients and medical personnel, as well as polluting surface or underground water bodies [16,17,18,21]. This problem of HPW has led to forever increasing levels of incineration, in an attempt to eliminate it, a fact which has been established by the statute of the national government (Colombia) [22]. According to the WHO, only state-of-the-art incinerators meet international standards for the emission of dioxins and furans from different sources, including the incineration of hospital waste [21,23]. Heat treatment of these wastes generates harmful, volatile and non-volatile organic compounds [16,22]. Among the compounds emitted are polycyclic aromatic hydrocarbons (PAH), polychlorinated dibenzo-p-dioxins (PCDD), and polychlorinated dibenzofurans (PCDF), which have been determined to be carcinogenic to humans by the International Agency for Research on Cancer. (IARC-WHO) [7,16,19,24,25].

The generation of these multiple pollutants creates the need to apply cleaner technologies, and one of them is pyrolysis, which provides the potential for energy and fuel recovery [26,27,28,29,30,31]. Pyrolysis has been applied to plastic mixtures [16,27,28,32,33,34], which makes it possible to recover high-value compounds [35,36,37] as well as gasoline [33] or diesel [26] range fuels. Pyrolysis then becomes a vitally important tool for the transformation of HRW into raw materials with high added value [33], and these values depend on the compositions of the hydrocarbons of interest, which are commonly analyzed by analytical techniques such as GC/MS [11,30,38,39,40]. The hydrocarbon profiles obtained are directly related to the proportions of each family of polymers present in the HPW, and these can be PE, PP, and PS, among others [41]. The many, varied properties of PE, PP and PS mixtures are influenced by the position and characteristics of the monomers, their stereochemistry, the types of intramolecular and intermolecular forces, the degree of symmetry and uniformity of the molecular structure, and many other factors [42,43]. This could lead to the occurrence of inter-molecular or extra-molecular reactions between these polymers during pyrolysis and thus generate several additional chemical compounds. This has not yet been evaluated with the current conventional technologies or used for the thermochemical processing of plastics/polymers [41,44,45,46].

Figure 1 describes an environmental management process that seeks to apply circular economic concepts of HRW recovery that do not involve specific filters, and we will seek to characterize this type of waste and provide a tool for its possible recovery. This study aims to determine a methodology for processing hospital plastic waste generated in five hospitals in the city of Cartagena-Colombia, applying the pyrolysis technique with a zero-emissions standard. TG-GC/MS is used to characterize the profile of molecules, quantify them and propose reaction mechanisms that allow us to understand their formation and be able to relate them to raw materials from the chemical, petrochemical, and polymer manufacturing.

## 2. Results and Discussion

### 2.1. Elemental Analysis

We determined the percentage composition of moisture, volatile material, and carbon and quantified the percentage by weight of the chemical elements that made up the different samples. The percentages of water, volatile substances, and carbon were established in the dry samples. Table 1 shows the results. With regard to humidity, it was found that the samples with the highest content were those from hospital 1 (0.45%) and hospital 5 (0.48%). For volatiles, the values were quite close to each other, and it was found that most of the composition of the hospital samples was of this type, as well as the humidity. The samples with the highest volatile values correspond to hospital 1 (99.65%) and hospital 4 (99.73%). Regarding the importance of residual matter (carbon), the highest values were for hospitals 2 (0.26%) and 3 (0.29%), respectively.

With regard to the elemental composition, it was found that the elements present in the analyzed samples were carbon, nitrogen, hydrogen and chlorine. For carbon, the samples with the highest concentrations by weight were hospital samples 2 (88.74%) and 4 (88.53%), respectively, and for hydrogen they were hospital samples 1 (6.34%) and 3 (7.23%). Nitrogen was hospital sample 3 (1.23%), and chlorine was hospital sample 4 (0.16%) and 5 (0.28%). All samples were shown to have a higher percentage of carbon. The lower calorific value (LHV) for all samples was higher than 40 MJ/Kg, and hospital 5, with 50.31 MJ/Kg, was the one with the highest registration, being higher than hospital 4 by 11.28%, hospital 3 by 6.34%, hospital 2 by 20.13% and hospital 1 by 11.89%.

**Table 1 ijms-24-05397-t001:** General composition of products obtained for the elemental analysis and polymeric composition as a percentage of HPW.

Analysis		HPW Hospital 1	HPW Hospital 2	HPW Hospital 3	HPW Hospital 4	HPW Hospital 5
Proximate analyses/wt.% (air-dried)	Moisture	0.45	0.27	0.35	0.22	0.48
	Volatile	99.65	99.47	99.36	99.73	99.45
	Fixed Carbon	0.1	0.26	0.29	0.05	0.07
Ultimate analyses/wt.% (dry ash-free)	Charcoal	85.32	88.74	82.38	88.53	85.93
	Hydrogen	6.346	5.651	7.235	5.127	4.932
	Nitrogen	0.24	0.35	1.23	0.53	0.25
	Chlorine	0.124	0.046	0.156	0.163	0.279
LHV	Lower Heating Value (MJ/Kg)	44.96	40.18	47.31	45.21	50.31
Composition analyses/wt.%	PP	86.46	84.25	85.41	84.53	86.19
	PS	12.23	11.95	12.73	11.71	11.65
	PE	2.31	3.8	1.86	3.76	2.16

### 2.2. Thermal Analysis

In the thermal analysis, we established that the samples were mainly composed of three polymers, which are polypropylene (PP), polystyrene (PS) and polyethylene (PE). The analysis values were established in percentage by weight, and it was found that the highest percentage of PP was in hospitals 1 (86.46%) and 5 (86.19%), respectively. The highest PS content was identified in hospitals 1 (12.23%) and 5 (12.73%). With regard to the EP, the samples of hospitals 2 (3.8%) and 4 (3.76%) were the highest. In general, given the three polymers identified in the samples, the polymer with the highest concentration in most samples was PP. All of the above information is presented in detail in Table 2. The organic composition of these residues is shown in Table 3, Table 4 and Table 5.

#### 2.2.1. ANOVA Analysis for PP, PS, and PR

In Table 3, based on Tukey’s analysis method and with 95% confidence, we can see that all mean values of PP, PE, and PS independently for each polymer in the five hospitals receive literal distinction (A, B, C) for their grouping. This indicates that, statistically, there are significant differences between these mean values. This statement is corroborated by the graph of Box chart for PP, PS, and PE, shown in Figure 2.

#### 2.2.2. ANOVA Analysis for Hospitals

In Table 3 based on Tukey’s analysis method and with 95% confidence, we can see that all the mean values of distribution in the composition of PP, PE, and PS per sample, in each of the hospitals receive the same literal (A) for their grouping. This indicates that, statistically, there are no significant differences between these mean values. This statement is corroborated by the hospital box diagram, shown in Figure 3.

#### 2.2.3. TGA

Figure 4a shows the TGA graph of pyrolysis of the samples of all hospitals superimposed on a ramp of 30 °C/min. It shows that the percentages are almost equal in terms of the distribution of the composition (PP, PS, and PE) of the samples of the five hospitals. The exact behavior of thermal degradation is presented, which suggests that these differences in composition are not big enough to be perceived, shown in the ANOVA analysis of Table 6, which generates changes that are not particularly remarkable in this decomposition. For the samples under these conditions, the range of most significant decomposition (HDG) was between 400 °C and 480 °C, where there was a decrease in the percentage by mass of 94.64% to 0.45%, respectively, which represents a percentage variation of the mass of 94.19%.

Figure 4b does not show the degradation behavior of samples in the oxidative atmosphere in the presence of oxygen and helium with the same heating ramp (30 °C/min) as pyrolysis. At a specific level, we observed that, as in pyrolysis, the thermal decomposition of each of the samples from the different hospitals had the same degradation behavior. At a general level, comparing pyrolysis with thermal oxidation (TO) shows that, in the latter, the percentage by mass of the samples presents a faster decrease depending on temperature, which is explained by the presence of oxygen, which generates more volatile species at a lower temperature, resulting in an earlier percentage decrease in the TGA curve comparing pyrolysis with OT both at a ramp of 30 °C/min. It was found that the latter presents a mass percentage reduction of 31.24% in a range of 210 to 410 °C, while pyrolysis presents a decrease of 8.6% in the same range.

#### 2.2.4. TG-GC/MS

##### The Gas Evolution Profiles during Medical Plastic Waste Thermal Degradation (Pyrolysis)

In the pyrolysis of the samples of the five hospitals, we identified with certainty and clarity a total of 26 compounds, 24 organic and 2 inorganics, categorized into aliphatic hydrocarbons, aromatic hydrocarbons and permanent gases. Table 3 shows the species identified by ion fragment, with their respective HRW value. It was established that, on average, 68.62% of hydrocarbons were aliphatic. Only two structural isomers of function were found. The hydrocarbons with oxygenated functions were 15.38% of aliphatic hydrocarbons and 8.33% of the total organic compounds found. In general, for all the samples analyzed from the five hospitals, the aliphatic hydrocarbon with the highest presence in the samples was propylene, with an average of 31.42%, and ethane, with an average of 8.81%. Aromatic hydrocarbons were, on average, 26.18% of the total compounds, of which 18.18% were aromatic hydrocarbons, and 8.33% were hydrocarbons with oxygenated functions. In the case of aromatic hydrocarbons, there was no substance with a majority of unanimous presence as in the case of aliphatic hydrocarbons. For the samples of hospitals 1, 3, and 4, the species with the highest amount was styrene, with an average of 14.59%; for the samples of hospitals number 2 and 5, it was toluene, with an average of 11.98%. Inorganic species coincided with the classification of permanent gases, despite having carbon atoms in their structure. Table 5 shows the percentage of gaseous species released from HRW thermal degradation in an inert atmosphere. Figure 5 shows a bar chart giving an overview of the products obtained by pyrolysis and that one of the most notorious aspects of this process is the almost total absence of gaseous products.

**Table 3 ijms-24-05397-t003:** Chemical species and their ion fragments obtained by TC-GC/MS.

Compounds	Detected Ion Fragments of *m*/*z*
1,3-butadiyne(C_4_H_2_), vinylacetylene(C_4_H_4_), 1,3-butadiene (C_4_H_6_) And 1-butene/2-Methylpropene (C_4_H_8_)	50–54, 56 and 57
cyclopentadienyl radical (C_5_H_5_), cyclopentane/methyl butene (C_5_H_10_) and pentane (C_5_H_12_)	65, 70, and 72, 62, 63, 67–69 and 71
(C_6_H_6_) and toluene (C_7_H_8_)	73, 74, 79–84 and 86. 78 and 92. 77 and 91
Styrene	103, 104
Ethylbenzene	103–106
Indene/α-methyl styrene	115, 116
Naphthalene	124, 126–128
Water (H_2_O), oxygen (O_2_), carbon monoxide (CO) and carbon dioxide (CO_2_)	18, 32, 28 and 44
Acetic acid	59, 60
propionic acid	59, 60, 73, 74
Phenol	93, 94
O-benzoquinone	107, 108
Benzoic acid	107, 108, 111, 112

**Table 4 ijms-24-05397-t004:** Results of pyrolysis of hospital plastic waste.

RT	Compounds	HPW Hospital 1	HPW Hospital 2	HPW Hospital 3	HPW Hospital 4	HPW Hospital 5
		He (RH%)	He (RH%)	He (RH%)	He (RH%)	He (RH%)
**Aliphatic hydrocarbons (not aromatic)**						
0.3	Methane (CH_4_)	1.86	1.45	2.01	2.43	5.79
2.1	Ethane (C_2_H_4_)	11.23	6.64	13.15	6.25	6.8
2.4	Ethylene (C_2_H_4_)	5.21	4.83	6.82	5.4	4.3
2.95	Propane (C_3_H_8_)	2.61	1.65	3.21	2.43	4.62
4.32	Vinylacetylene (C_4_H_4_)	2.86	2.45	4.26	2.57	1.38
5.48	Propylene (C_3_H_7_)	12.42	16.45	37.47	40.5	50.27
7.35	Pentane (C_5_H_12_)	0.42	1.25	0.94	1.02	1.3
8.98	1-Butene (C_4_H_8_)	3.11	2.42	4.16	2.6	1.73
11.26	Methyl butene (C_5_H_10_)	0.75	2.86	1.23	1.12	0.88
13.08	1-Propanol (C_3_H_8_O)	0.2	1.57	0.43	1.36	2.33
14.99	Ethyne (C_2_H_2_)	2.03	3.42	2.63	2.03	1.53
15.36	Acetic acid (C_2_H_4_O_2_)	0.3	1.52	1.03	1.16	2.76
15.86	1.3-butadiene (C_4_H_6_)	3.11	4.83	4.25	3.72	3.45
**% Total aliphatic hydrocarbons**		**46.83**	**52.89**	**81.71**	**73.51**	**88.19**
**Aromatic hydrocarbons**						
12.84	Benzene (C_6_H_6_)	0.72	1.55	0.12	0.92	1.05
16.59	Toluene (C_7_H_8_)	15.62	19.43	5.13	8.56	4.52
17.45	Ethylbenzene (C_8_H_10_)	5.11	8.22	1.43	0.4	0.95
18.72	Phenol (C_6_H_6_O)	0.53	1.52	0.02	0.01	0.11
19.61	Benzene. 1.3-dimethyl (C_8_H_10_)	3.35	2.44	0.86	1.32	0.76
20.34	Styrene (C_8_H_8_)	22.43	11.72	9.13	12.2	1.24
21.16	Benzoic acid (C_7_H_6_O_2_)	0.36	0.15	0.01	0.43	0.73
21.98	Indene (C_9_H_8_)	1.54	0.83	0.24	0.83	0.16
22.74	α-methyl styrene (C_9_H_10_)	0.73	0.23	0.05	0.01	0.05
23.52	Hydrindene (C_9_H_10_)	0.28	0.13	0	0	0
24.38	Naphthalene (C_10_H_8_)	2.86	1.53	0.2	1.23	1.38
**% Total Aromatic hydrocarbons**		**52.81**	**46.20**	**17.02**	**24.99**	**9.90**
**Permanent gases**						
	Carbon dioxide (CO_2_)	0.01	0.02	0.01	0.03	0.04
	Carbon Monoxide (CO)	0.05	0.01	0.01	0.01	0.02
**% Total permanent gases**		**0.06**	**0.03**	**0.02**	**0.04**	**0.06**
Other		0.3	0.88	1.2	1.51	1.85

**Figure 5 ijms-24-05397-f005:**
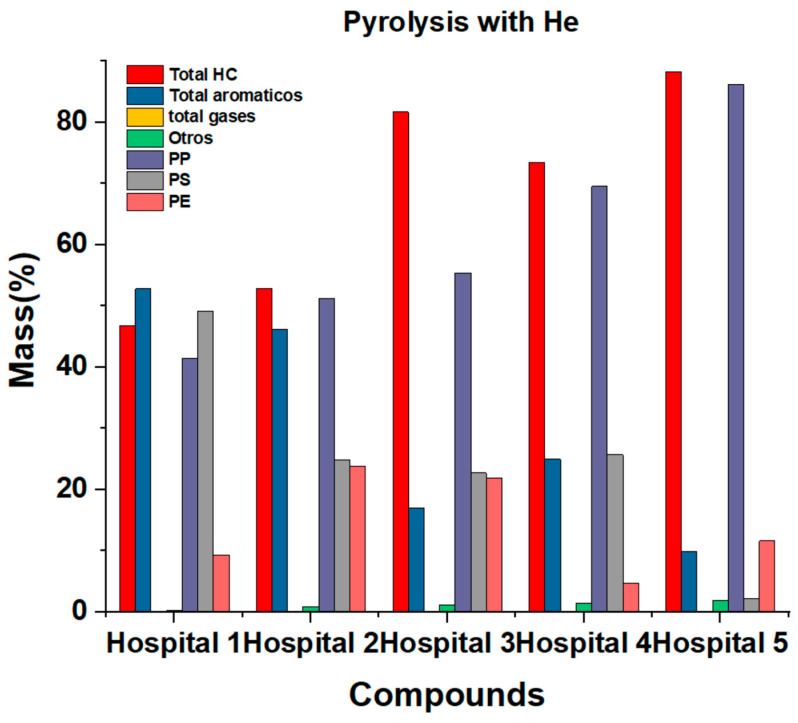
Pyrolysis result bar diagram.

##### The Gas Evolution Profiles during Medical Plastic Waste Thermal Degradation (Oxidative Atmosphere)

As for the inert atmosphere (pyrolysis), for the oxidative atmosphere (TO), the amounts obtained as a percentage for the same 26 species and under the same categorization were analyzed (Table 5). However, we received markedly different results for degradation in the oxidative atmosphere. Under these conditions, there was a more significant number of gaseous products and the total absence of certain species in the samples of some hospitals and partial absence (lack in some samples and not in others) of some other species that we obtained in the pyrolysis of the HRW.

**Table 5 ijms-24-05397-t005:** Results of oxidative decomposition of hospital plastic waste.

**RT**	**Compounds**	**HPW** **Hospital 1**	**HPW** **Hospital 2**	**HPW** **Hospital 3**	**HPW** **Hospital 4**	**HPW** **Hospital 5**
		**O_2_/He**	**O_2_/He**	**O_2_/He**	**O_2_/He**	**O_2_/He**
**Aliphatic hydrocarbons (not aromatic)**						
0.3	Methane (CH_4_)	0.05	0.1	0.2	0	0
2.1	Ethane (C_2_H_6_)	1.95	0.1	3.12	2.87	1.32
2.4	Ethylene (C_2_H_4_)	0	0	0	0	0
2.95	Propane (C_3_H_8_)	0	0.31	0	0	0
4.32	Vinylacetylene (C_4_H_4_)	0	0	0	0	0
5.48	Propylene (C_3_H_7_)	1.24	2.1	3.21	4.23	4.78
7.35	Pentane (C_5_H_12_)	0	0	0	0	0
8.98	1-Butene (C_4_H_8_)	0	0	0	0	0
11.26	Methyl butene (C_5_H_10_)	0	0	0	0	0
13.08	1-Propanol (C_3_H_8_O)	0	0	0	0	0
14.99	Ethyne (C_2_H_2_)	0.56	0	0.93	0.36	0.17
15.36	Acetic acid (C_2_H_4_O_2_)	1.54	2.31	3.12	3.91	4.27
15.86	1.3-butadiene (C_4_H_6_)	0.1	1.1	0	0	0
**% Total aliphatic hydrocarbons**		**5.44**	**6.02**	**10.58**	**11.37**	**10.54**
**Aromatic hydrocarbons**						
12.84	Benzene (C_6_H_6_)	0	0	0	0	0
16.59	Toluene (C_7_H_8_)	2.42	3.45	0.62	0.72	0.31
17.45	Ethylbenzene (C_8_H_10_)	0.01	0.1	0	0	0
18.72	Phenol (C_6_H_6_O)	1.43	2.3	0	0	0
19.61	Benzene. 1.3-dimethyl (C_8_H_10_)	0.02	0	0	0	0
20.34	Styrene (C_8_H_8_)	6.87	8.25	0.73	0.96	0.12
21.16	Benzoic acid (C_7_H_6_O_2_)	1.35	0	0	0	0
21.98	Indene (C_9_H_8_)	0	0	0	0	0
22.74	α-methyl styrene (C_9_H_10_)	0.03	0	0	0	0
23.52	Hydrindene (C_9_H_10_)	0.01	0	0	0	0
24.38	Naphthalene (C_10_H_8_)	0.06	0	0	0	0
**% Total Aromatic hydrocarbons**		**12.20**	**14.10**	**1.35**	**1.68**	**0.43**
**Permanent gases**						
	Carbon dioxide (CO_2_)	63.87	72.45	85.28	82.88	84.94
	Carbon Monoxide (CO)	11.75	7.43	2.57	3.54	3.79
**% Total permanent gases**		**75.62**	**79.88**	**87.85**	**86.42**	**88.73**
Other		6.74	0	0.21	0.53	0.35

For this degradation, the species we found in smaller quantities were aliphatic hydrocarbons, with an average of 8.79%, followed by aromatic hydrocarbons, with an average of 5.95%. On the other hand, most of the products were gaseous species, mostly permanent gases that presented an average emission of 83.7%. Some species that showed total absence in aliphatic hydrocarbons were ethylene, vinyl acetylene, pentane, 1-butane, methyl butane, and 1-propanol. For aromatic hydrocarbons, the completely absent species were benzene and indene. Figure 6 shows the species released from HPW thermal degradation in an oxidative atmosphere of O_2_/He. There is also an almost proportional increase in hospital polypropylene 1 to 5, almost as if the samples were correlated (Figure 5 and Figure 6). Figure 7 shows the distribution of the data by family and by compounds.

#### 2.2.5. ANOVA TGA Pyrolysis Analysis vs. Oxidative Atmosphere

It should be clarified that all the graphs and tables shown below are based on Tukey’s analysis method, with a degree of confidence of 95%. First, Table 6 shows the ANOVA for aliphatic hydrocarbons (ALH) resulting from pyrolysis (inert atmosphere of He), For Figure 8, according to Tukey’s table, we state that, except for propylene, the other substances share the same literal (B), which means that there are no statistically significant differences between them. However, comparing the above with Table 6, we see that acetic acid and ethane change from literal, sharing it with propylene, so for these three ALH in the oxidative atmosphere, there is no relevant significance in their differences, although, concerning the rest of ALH, the rest of the hydrocarbons did not have an applicable difference between them even when the presence of certain species was null. For this comparison in pyrolysis, 92.3% of the HLA did not show significant differences, while for thermal oxidation, it was 76.9%. At this point, it is valid to clarify that all Tukey’s analysis tables are accompanied by a graph of boxes that allows the visual corroboration of the information in the tables and facilitates an understanding of these.

**Table 6 ijms-24-05397-t006:** ANOVA analysis for families of products obtained in different atmospheres.

Factor	N	Average	Grouping
**Aliphatic Inert Atmosphere (He)**			
Propylene (C_3_H_7_)	5	31.42	A
Ethane (C_2_H_4_)	5	8.81	B
Ethylene (C_2_H_4_)	5	5.312	B
1,3-butadiene (C_4_H_6_)	5	3.872	B
Propane (C_3_H_8_)	5	2.904	B
1-Butene (C_4_H_8_)	5	2.804	B
Methane (CH_4_)	5	2.708	B
Vinylacetylene (C_4_H_4_)	5	2.704	B
Ethyne (C_2_H_2_)	5	2.328	B
Methyl butene (C_5_H_10_)	5	1.368	B
Acetic acid (C_2_H_4_O_2_)	5	1.354	B
1-Propanol (C_3_H_8_O)	5	1.178	B
Pentane (C_5_H_12_)	5	0.986	B
**Factor**	**N**	**Average**	**Grouping**
**Aliphatic oxidative atmosphere (O_2_/He)**			
Propylene (C_3_H_7_)	5	3.112	A
Ethane (C_2_H_6_)	5	3.030	A
Ethylene (C_2_H_4_)	5	1.872	A
1,3-butadiene (C_4_H_6_)	5	0.404	B
Propane (C_3_H_8_)	5	0.240	B
1-Butene (C_4_H_8_)	5	0.0700	B
Methane (CH_4_)	5	0.0620	B
Vinylacetylene (C_4_H_4_)	5	0.0	B
Ethyne (C_2_H_2_)	5	0.0	B
Methyl butene (C_5_H_10_)	5	0.0	B
Acetic acid (C_2_H_4_O_2_)	5	0.0	B
1-Propanol (C_3_H_8_O)	5	0.0	B
Pentane (C_5_H_12_)	5	0.0	B
**Factor**	**N**	**Average**	**Grouping**
**Aromatic inert atmosphere (He)**			
Styrene (C_8_H_8_)	5	11.34	A
Toluene (C_7_H_8_)	5	10.65	A
Ethylbenzene (C_8_H_10_)	5	3.22	B
Benzene, 1,3-dimethyl (C_8_H_10_)	5	1.746	B
Naphthalene (C_10_H_8_)	5	1.440	B
Benzene (C_6_H_6_)	5	0.872	B
Indene (C_9_H_8_)	5	0.720	B
Phenol (C_6_H_6_O)	5	0.438	B
Benzoic acid (C_7_H_6_O_2_)	5	0.336	B
α-methyl styrene (C_9_H_10_)	5	0.214	B
Hydrindene (C_9_H_10_)	5	0.0820	B
**Factor**	**N**	**Average**	**Grouping**
**Aromatic Oxidative atmosphere (O_2_/He)**			
Styrene (C_8_H_8_)	5	3.39	A
Toluene (C_7_H_8_)	5	1.504	A B
Ethylbenzene (C_8_H_10_)	5	0.746	A B
Benzene, 1,3-dimethyl (C_8_H_10_)	5	0.270	B
Naphthalene (C_10_H_8_)	5	0.0220	B
Benzene (C_6_H_6_)	5	0.0120	B
Indene (C_9_H_8_)	5	0.00600	B
Phenol (C_6_H_6_O)	5	0.00400	B
Benzoic acid (C_7_H_6_O_2_)	5	0.00200	B
α-methyl styrene (C_9_H_10_)	5	0.0	B
Hydrindene (C_9_H_10_)	5	0.0	B
**Factor**	**N**	**Average**	**Grouping**
**Permanent gases Inert atmosphere (He)**			
Carbon dioxide (CO_2_)	5	0.02200	A
Carbon monoxide (CO)	5	0.02000	A
**Factor**	**N**	**Average**	**Grouping**
**Permanent gases Oxidative atmosphere (O_2_/He)**			
Carbon dioxide (CO_2_)	5	77.88	A
Carbon monoxide (CO)	5	5.82	B

The same comparison is made in the second instance for aromatic hydrocarbons (ARH) resulting from the thermal degradation processes. In Table 6, concerning the ARH of pyrolysis, we find that styrene and toluene are the highest means without a significant difference between them, but the rest of ARH of pyrolysis and that of ARH do not have a considerable difference between them. However, for Table 6, concerning the ARH of thermal oxidation, notoriously different results were obtained since toluene and phenol are intercepted in the literals of group A and B, which shows that these two species do not have significant differences concerning the ARH that belongs to literal A (styrene) but nor do they have a considerable difference concerning the ARH of literal B. They do not even present significant differences between them. For this comparison in pyrolysis, 84.6% of the HLA showed no difference in significance, while for thermal oxidation, it was 92.3% (literal A), with 15.3% that presented no difference concerning any of the groups (literal A and B).

In the case of permanent gases PG, in pyrolysis, there were no significant differences, and their percentage amounts were low (≤0.06), which is shown in Table 6. However, for thermal oxidation there were very significative differences, since we only generated more by the PG if the difference between their quantities was quite comprehensive (92.5%), which we can see in Table 6 by observing the value of their means. Thus, it is demonstrated that the processes of pyrolysis and thermal oxidation generate significantly different results in terms of the presence of species, their quantities, and the significant differences between them.

### 2.3. Possible Reaction Mechanism for the Species Found

#### 2.3.1. Possible Reaction Mechanism in Pyrolysis of HRW and Medical Remains of Triglycerides (TMT)

When HPW and triglyceride medical remains (TMT) are pyrolyzed, primary and secondary cracking are expected to be the two main reactions. The cleavage of the connection between the carbon chain and oxygen causes the direct cracking of the triglyceride molecules TMT at low degradation temperatures (300–350 °C), which connects glycerol and fatty acids. The carboxyl group is then released, and this removal transforms the organic acids present in the TMTs and generates carboxylate radicals [47,48]. The rupture of the structure of the polymers in HPW could have released hydrogen-free radicals when the temperature rose to 380 °C. These radicals are chemically reactive compounds containing unpaired electrons. During pyrolysis, radicals are created by substitution, addition, or removal processes. In TMTs, oxygen-containing molecules, such as fatty acids, and double-bonded compounds, such as unsaturated fatty acids and alkenes, are likely to break down due to interactions and integration between HPW hydrogen-free radicals and RCOO-radicals (Figure 8). As a result, the production of oxygen-containing chemicals, such as carboxylic acids and alcohols, was reduced, while propylene, styrene, ethane, and toluene production increased. Due to the application of TMT/HPW as pyrolysis samples, the alkanes’ composition in the obtained products changed drastically. HPW and TMT alkenes break at elevated decomposition temperatures (400–480 °C, Figure 4), resulting in the generation of light alkanes (e.g., C_1_–C_3_).

The reduced composition of organic acids in the chemical species obtained by pyrolysis compared to those obtained by TMT degradation is only explained by the fact that certain organic substances possessing functional groups such as carbonyl and carboxyl crack at higher levels to form new organic substances (hydrocarbons) during the secondary cracking stage. The interactivity between olefins and the species formed from the breaking of bonds of glycerin found in TMTs by the reaction of the addition of Diels-Alder ethylene in the smelting of HPW and TMT may be the source of cyclic poly-alkenes such as benzene and related species (Figure 8) [48].

#### 2.3.2. Possible Reaction Mechanism in Pyrolysis of HRW and Medical Remains of Wood or MWW/P Paper (Biopolymers [Lignin, Cellulose and Hemicellulose])

No alkenes were discovered among the products generated by pyrolysis of HPW/(MWW/P) at temperatures below 353 °C, because it is presumably too low to start olefins hydrogen addition. This justifies the absence of saturated hydrocarbons (alkanes). At this temperature, however, the transformation of alkenes into alkanes at high temperatures (400–480 °C, Figure 4) during pyrolysis is appreciated.

Dehydration of cellulose causes certain oxygenates to turn into the water by removing the hydroxyl group from the cellulose. In addition, cellulose dehydration produces hydroxyl radicals, which interact with HPW hydrocarbons to form benzene rings containing OH groups as a substituent, known as phenols, by polymerization. Likewise, the deformation of the lignin structure could cause the breaking of ether bonds, eliminating oxygenation with aldehyde, carboxyl, and hydroxyl groups of lignin (Figure 9) [48,49,50,51,52,53,54,55].

When cellulose and hemicellulose break down, oxygenates can be produced, including alcohols. These oxygenate induce cracking events, water loss, and the formation of polymers that transform oxygenated species in Benzene derivatives when they come into contact with HPW hydrogen-free radicals. The subsequent decomposition and deformation of structures containing benzene rings are combined with functional groups, resulting in benzene derivatives containing hydroxyl groups (e.g., Phenols) [49]. According to a series of processes, such as the formation of oligomers, the loss of the carbonyl group, and the loss of the carboxyl group, they produce organic molecules with hydroxyl (-OH) groups, which connect to benzene rings and others, to generate phenols and others (Figure 9).

#### 2.3.3. Possible Reaction Mechanism in Pyrolysis of HRW for Polystyrene PS

Most HRWs are composed of PS and PP, generated using the polymerization of various monomers. Figure 8 describes a comprehensive method for the thermal deterioration of PS. The reactions that cause the degradation of PS are [50,51]: (A) Arbitrary division (See Figure 10), involving the formation of free radicals and producing resulting structures of different extensions, creating primary and secondary hierarchical macro-radicals. The two macro-radicals will later be combined to generate a collection of compounds that will be value-added products of interest to different industries; (B) Depolymerization (Process 2, Figure 10), occurring as a function of the low-strength bonds, which causes the PS to decompose into styrene plus a radical detached from the type of plastic in question. Styrene plus propagation-independent radicals can be produced at the base when a polymer undergoes cleavage due to the presence of free radicals [50], which also demonstrates TG-GC/MS; (C) Intramolecular transfer (Process 3 and 4, Figure 7b), intramolecular transfer (Processes 3 and 4, Figure 10), which involves the intramolecular transfer of C1–C3 and C1–C5 from the outer group of the secondary macroradical belonging to the PS specifically tertiary carbon (reactive) and the ß-cleavage, will possibly lead to the composition of a variety of degradation products, distinguishing it from 2 (dimer) and 3 (trimer) [52], units of the PS monomer (styrene). This, in turn, leads to aromatic carbon-carbon double bonds and single aromatic carbon-hydrogen bond groups. TG-GC/MS was used to identify PS monomer (styrene), benzene, toluene, and certain C1–C4 unsaturated hydrocarbon species.

#### 2.3.4. Possible Reaction Mechanism in Pyrolysis of HRW for Polystyrene PS

Various studies have been conducted on the early thermal decomposition of PS, PP, and PS/PP mixtures [50,52,53,54]. However, the secondary chemical combinations of the first thermally degraded species generated from the pyrolysis of medical plastic waste are rarely documented. The key volatiles generated from the decomposition of medical plastic waste are represented in Figure 11, which are deduced from the TG-GC/MS results.

According to TG-GC/MS data, the main gaseous products emitted by pyrolysis of medical plastic waste are ethylene (C_2_H_4_), benzene (C_6_H_6_), toluene (C_7_H_8_), styrene (C_8_H_8_) and ethylbenzene (C_8_H_10_). According to [50], olefins such as ethylene (C_2_H_4_) and propylene (C_3_H_6_) can go through a sequence of possible side reactions to produce propane (C_3_H_8_, mass/load = 44), 1,3-butadiene (C_4_H_6_, mass/load = 53, 54), vinylacetylene (C_4_H_4_, mass/load = 51, 52), and 1-butene (C_4_ By cracking PP, pentane (C_5_H_12_) and methylbutene (C_5_H_10_) are likely to be produced. The combinations of ethylene (C_2_H_4_) and propylene (C_3_H_6_) with O result in the production of ethanoic acid (C_2_H_4_O_2_, mass/load = 59, 60) [52]. After this, benzene is abbreviated as B_1_. During the pyrolysis of PS, one of the primary volatile products is styrene (B_1_C_2_H_3_, mass/load = 104) [52]. Styrene (B_1_C_2_H_3_) is a monomer that decomposes monomolecularly to give benzene (A1, mass/load = 78) + C_2_H_2_, cyclopentadienyl radical (C_5_H_5_, mass/load = 65) + C_3_H_3_, and vinylacetylene (C_4_H_4_, mass/bag = 52) + vinylacetylene (C_4_H_4_, mass/gear = 52). GC/MS can also identify vinylacetylene and cyclopentadienyl. Styrene and OH should combine to produce phenol (B_1_OH, mass/charge = 94). However, at higher temperatures, styrene can also be made by combinations of vinylacetylene (C_4_H_4_, mass/charge = 52) and vinylacetylene (C_4_H_4_, mass/charge = 52), but can also be with the 1,3-butadienyl radical (C_4_H_5_, mass/charge = 53). The O atom and styrene can react in an oxidizing environment to form benzaldehyde (B_1_CHO, mass/load = 106), which can react with the O atom to create benzoic acid (B_1_COOH, mass/charge = 112) in an oxygen-rich atmosphere. The hydrogenation process of styrene is primarily responsible for forming ethylbenzene (B_1_C_2_H_5_, mass/load = 106) [54]. Likewise, ethylbenzene can be broken down into the phenyl radical (B_1_-, mass/gear = 77) + ethylene (C_2_H_4_, mass/load = 28) or styrene (B_1_C_2_H_3_, *m*/*z* = 104) by processes of abstraction of H atoms and unimolecular decomposition. Toluene (B_1_CH_3_, mass/load = 92) and H can react ipso-substitutively to produce the methyl radical (CH_3_) and benzene (B_1_, mass/load = 78). By replacing styrene methyl and crackling PS, α-methyl styrene is mainly generated [55,56]. Toluene (A_1_CH_3_, *m*/*z* = 92) and H can also substitutionally react to generate benzene (A_1_, *m*/*z* = 78) and the methyl radical (CH_3_) [57,58]. The methyl substitution of styrene and crackle PS generates mainly α-methyl styrene.

Toluene (B_1_CH_3_, mass/load = 92) and OH undergo a substitution reaction to an already substituted carbon of the aromatic ring in an oxidizing environment to form phenol (B_1_OH, mass/load = 94), as shown in Figure 11 and Figure 12. Toluene (B_1_CH_3_, mass/gear = 92) also produces benzaldehyde (B_1_CHO, mass/load = 106) when it reacts with oxygen, and this benzaldehyde can combine with another oxygen atom to generate phenylcarboxylic acid (B_1_COOH, mass/gear = 112) [59]. The degradation of HRW is usually due to the cracking of the styrene dimer. This generates benzene (B_1_, mass/load = 78) as a primary and intermediate product. In addition, Figure 11 shows numerous other pathways for synthesizing benzene, including C_2_ and C_4_, C_3_ and C_3_, and C_4_ and C_4_ reactions [59]. However, unimolecular disintegration reactions, isolation of the H atom, theft of the H atom, attack on substituted carbons, and even the O atom are likely to occur in benzene decay into various components. The essential benzene decomposition pathways, according to the TG-GC/MS, include O and OH attacking the O atom to produce phenol (B_1_OH, mass/load = 94) and o-benzoquinone (OA_1_O, mass/load = 108) in an oxidizing environment [60]. Meanwhile, the cyclopentadienyl radical (C_5_H_5_, mass/charge = 65) and the 1,3-cyclopentadiene (C_5_H_6_, mass/charge = 66) are expected to be produced during phenol decarbonylation (B_1_OH, mass/load = 94). The primary source of polycyclic aromatic hydrocarbons, such as naphthalene, phenanthrene, fluorene, and biphenyl [61], is benzene and its derivatives, which interact with each other. According to Figure 12, the predominant production routes of naphthalene (C_10_H_8_, masse/landing = 128) and indene (C_9_H_8_, masse/landing = 116), in which GC/MS recognizes coupled TG, are combinations of the phenyl or benzene radical with the vinylacetylene radical and the propargyl radical. Naphthalene (C_10_H_8_, mass/load = 128) can also be created by cyclopentadienyl self-combination (C_5_H_5_, mass/load = 65) since cyclopentadienyl is likewise detectable by GC/MS [62].

## 3. Materials and Methods

The samples of plastic waste collected from the five hospitals in the city of Cartagena-Colombia were composed especially of plastic bottles containing pharmaceutical waste and plastic bags used in infusion processes. The purpose or goal is to understand whether drugs and other substances inside containers could alter the thermal consistency of plastic waste. These residues were not subjected to a sterilization process to fulfill the purpose above; they were only subjected to drying, cutting, and grinding until particles 0.1 mm thick were obtained.

### 3.1. Apparatus and Procedure

#### 3.1.1. Elemental Analysis

The presence of metals and other chemical components in the samples is identified and quantified using Malvin’s Axios FAST X-ray fluorescence instrumental analysis technique. The low calorific measurements of the samples already processed (0.1 mm) were established by using a calorimetric pump (Model 1261 instruments for parr), according to ASTM D 5865-04.

#### 3.1.2. Humidity

For moisture analysis of HPW samples, a Vapor ProXL Computrac (BROOKFIELD AMETEK, 11 Commerce Blvd, Middleborough, MA 02346, USA) equipment was used. 5 g of HPW samples were weighed and deposited in a 25 mL glass vial.

#### 3.1.3. TG-GC/MS Experiment

Thermogravimetry (TG) (NETZSCH STA 449 F3) was also used in combination with gas chromatography, coupled with mass spectrometry (GC/MS) to perform pyrolysis/combustion tests. About 25 mg of sample with diameters less than 0.1 mm were pyrolyzed in the TG. Table 7 shows the conditions.

The first column shows the composition analysis of the substances with low vapor pressure and the second column shows a quantification of the concentrations of CO and CO_2_. The sensor device was a selective electron collision mass and mass spectrum detector with an ionizing energy value equal to 70 eV with a scanning range of 10 to 50 *m*/*z*. The MS transfer line and ion source temperature were preserved at 300 °C.

## 4. Conclusions

The sampling of medical waste from the five hospitals showed the presence of PP, PS, and PE in average compositions of 85.37, 12.05 and 2.78%, respectively. The average pyrolysis of the five residues allowed 31.42% propylene, 8.81% ethane, 30.2% aromatic hydrocarbons, and 0.04% permanent gases to be obtained. The aromatic compounds present in the highest concentration were toluene, ethylbenzene, styrene, 1,3-dimethylbenzene, indene and naphthalene. In addition, this hospital waste was degraded in an oxidative atmosphere with a mixture of O_2_/He, which allowed an average content of 8.79% aliphatic hydrocarbons, 5.92% aromatic hydrocarbons and 83.7% permanent gases to be obtained. The five routes of reaction mechanisms that were proposed allowed us to understand the formation of all chemical species with functional groups of alkanes, alkenes, carboxylic acids, alcohols, aromatics and permanent gases that were quantified in this investigation using mass spectrometry.

## Figures and Tables

**Figure 1 ijms-24-05397-f001:**
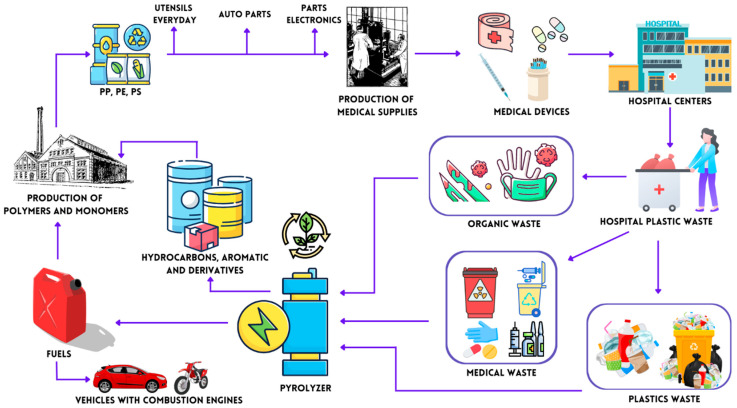
Flowchart for HRW recirculation process.

**Figure 2 ijms-24-05397-f002:**
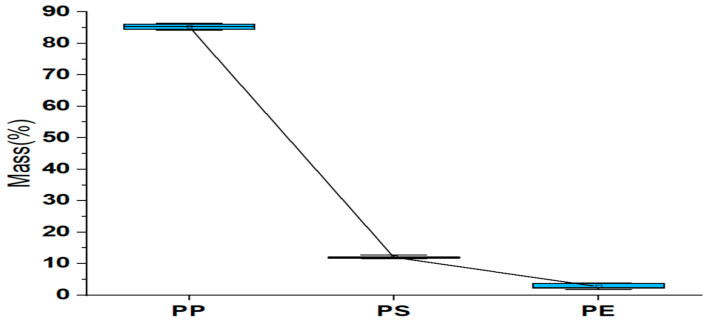
Box chart for PP, PS, and PE.

**Figure 3 ijms-24-05397-f003:**
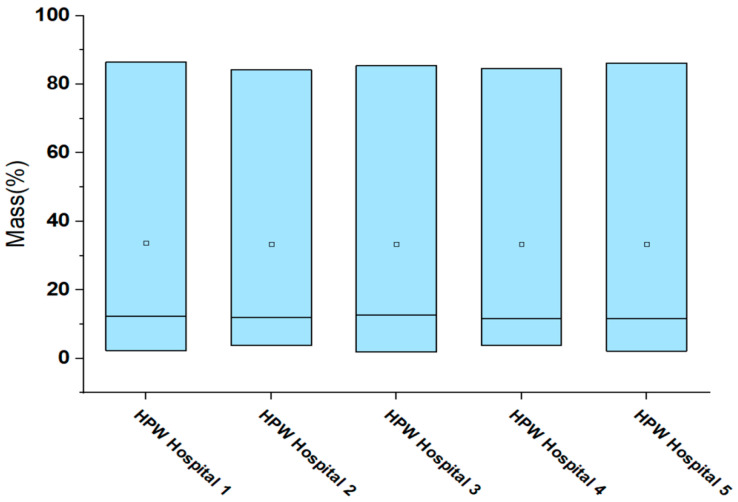
Box chart for Hospitals.

**Figure 4 ijms-24-05397-f004:**
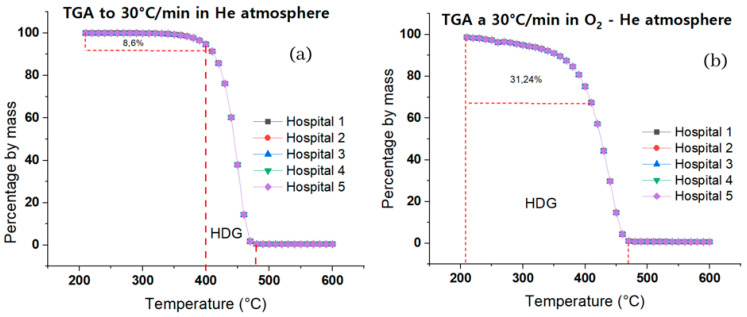
Thermal degradation behavior of pyrolysis. (**a**) He atmosphere (**b**) O_2_—He atmosphere.

**Figure 6 ijms-24-05397-f006:**
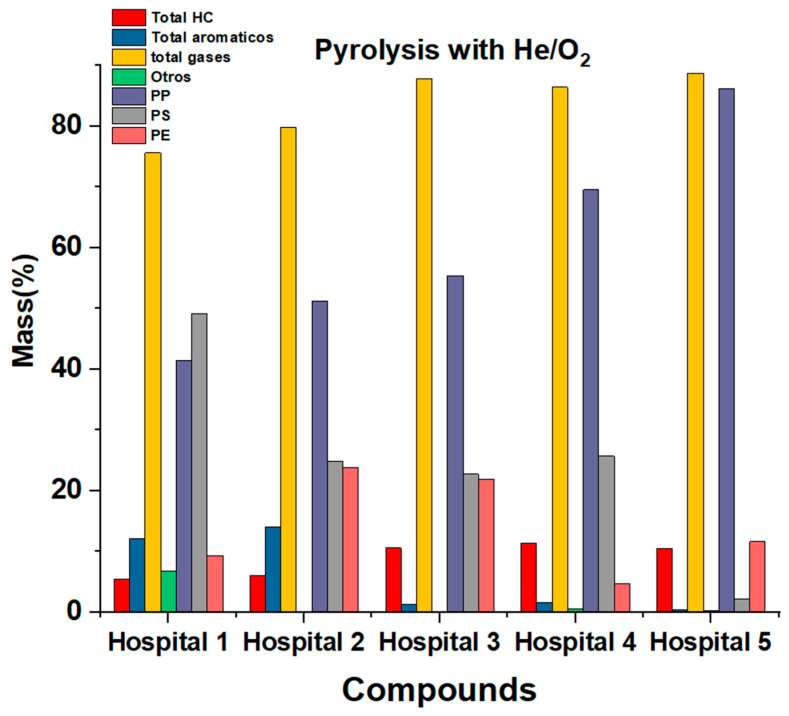
Bar diagram of oxidative decomposition result.

**Figure 7 ijms-24-05397-f007:**
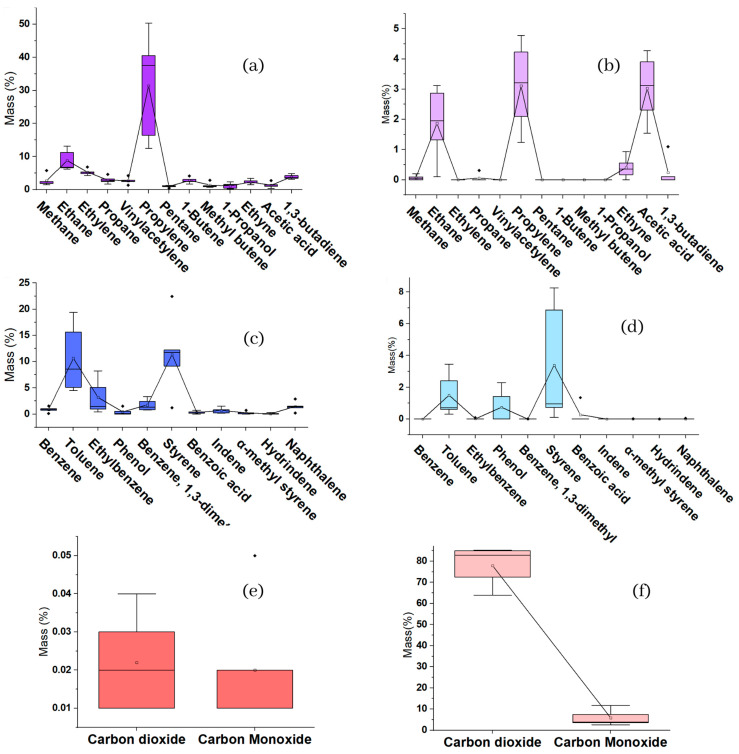
(**a**) Box plot of aliphatic inert atmosphere (He). (**b**) Box plot of aliphatic oxidative atmosphere (O_2_/He). (**c**) Box plot of aromatic inert atmosphere (He). (**d**) Box plot of aromatic oxidative atmosphere (O_2_/He). (**e**) Box plot of permanent gases Inert atmosphere (He). (**f**) Box plot of permanent gases oxidative atmosphere (O_2_/He).

**Figure 8 ijms-24-05397-f008:**
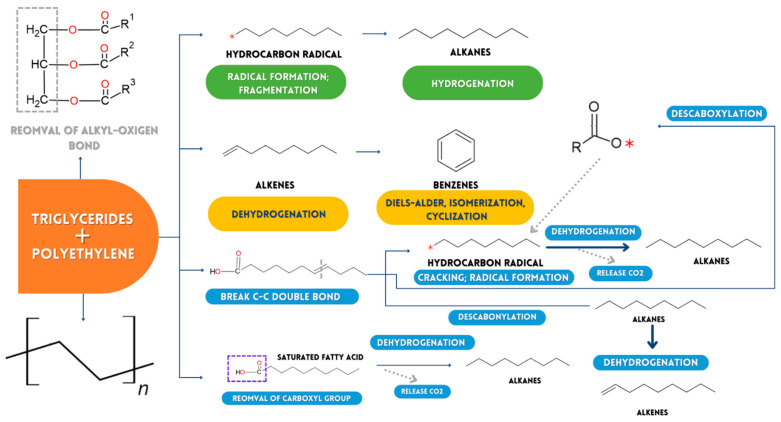
Possible flow chart of chemical decomposition of polyethylene and triglyceride $\left(*\right)\;Free\;radical$.

**Figure 9 ijms-24-05397-f009:**
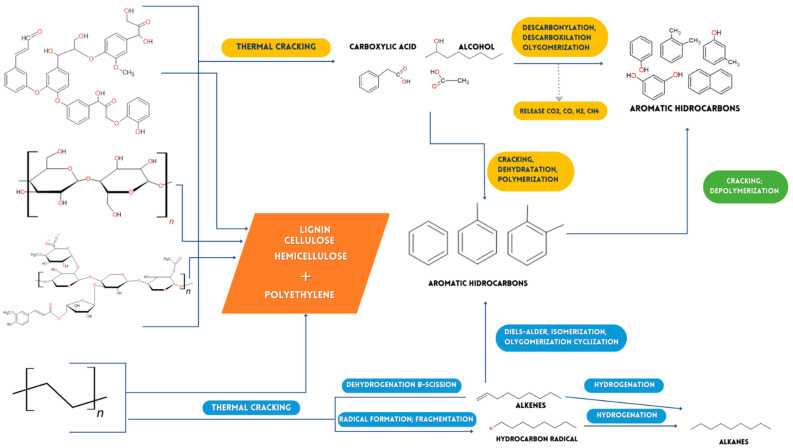
Possible flow of chemical decomposition of polyethylene, lignin, hemicellulose and cellulose.

**Figure 10 ijms-24-05397-f010:**
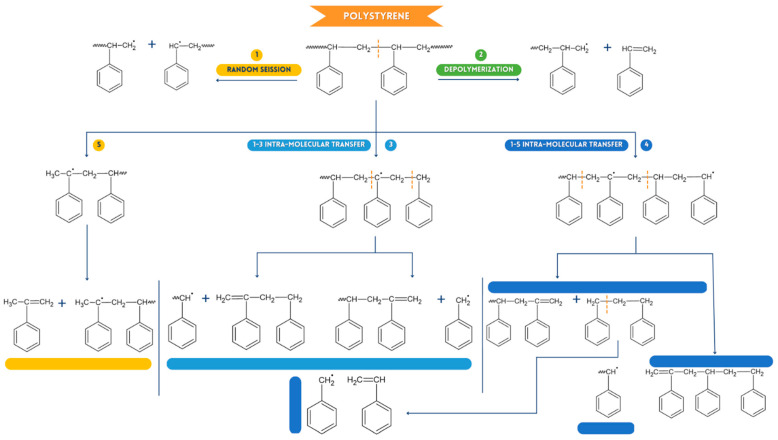
Possible flow of chemical decomposition of polystyrene.

**Figure 11 ijms-24-05397-f011:**
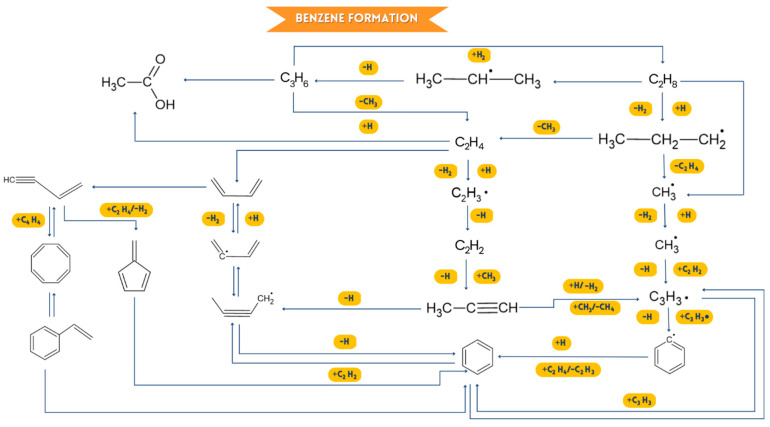
Possible flow of benzene formation flow.

**Figure 12 ijms-24-05397-f012:**
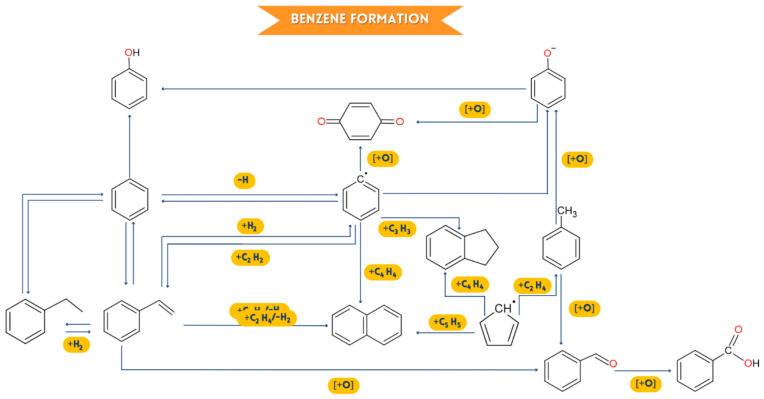
Possible flow of benzene formation flow (Other way).

**Table 2 ijms-24-05397-t002:** Statistical analysis of PP, PS, PE, and Hospitals. (Means that do not share a letter are significantly different).

Factor	N	Average	Grouping
ANOVA Analysis of PP, PS, and PE			
PP	5	85.368	A
PS	5	12.054	B
PE	5	2.778	C
**Factor**	**N**	**Average**	**Grouping**
**ANOVA Analysis for Hospitals**			
HPW Hospital 1	3	33.7	A
HPW Hospital 5	3	33.3	A
HPW Hospital 4	3	33.3	A
HPW Hospital 3	3	33.3	A
HPW Hospital 2	3	33.3	A

**Table 7 ijms-24-05397-t007:** Conditions of analysis of TG and the chromatograph.

TG	
Temperature	40–900 °C. 10 °C/min.
Flow	20 mL/min He (Mixture 16:4 He: O_2_ in oxidative atmosphere)
**Chromatography**	
Agilent 7820A-5975	**Column #1:** HP-5MS (5% Phenyl Methyl Silox, 30 m × 0.25 mm I.D. × 0.25 μm), **Column #2:** HP-MOLESIEVE (30 m length, 0.53 mm I.D)
He Carrier: 1 mL/min	Oven temperature: 40 °C × 4 min, then heated to 250 °C a 10 °C/min, and 250 °C × 5 min. split ratio is 10:1. S source and MS quad are 230 and 150 °C

## Data Availability

Not applicable.

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
