# Peer review of "Application of Pyrolysis for the Evaluation of Organic Compounds in Medical Plastic Waste Generated in the City of Cartagena-Colombia Applying TG-GC/MS"

_ijms, 2023, doi:10.3390/ijms24065397_

Round 1

Reviewer 1 Report

The introduction and conclusion sections should be rewritten in terms of fluent reading of the manuscript. For example in the introduction, "in addition" is used a lot, which makes it difficult to read fluently.

The writing mistakes that capital letters after the comma, lower case after a period should be corrected.

Graphics are needed to drawn with a scientific program, they all need to be drawn in the same format and with an appropriate scientific image, in some of them the name of the  y axis is not given clearly, which is weak for an article image

As a result, the manuscript is worth publishing in terms of subject and scope, but making these revisions are important in terms of article quality.

Author Response

Dear 

Thank you once again for your time invested in reviewing this research and for recommendations that have allowed improvements. We attach the answers.

1-The introduction and conclusion sections should be rewritten in terms of fluent reading of the manuscript. For example in the introduction, "in addition" is used a lot, which makes it difficult to read fluently.

R/ Good afternoon. Very kind of you to agree to review this research. Following his recommendation, the entire introduction and conclusions have been rewritten to ensure greater fluidity. English was also improved with the help of a native researcher.

2-The writing mistakes that capital letters after the comma, lower case after a period should be corrected.

R/ Good afternoon. Very kind of you for accepting the review of this research. The letters after the points have been corrected to capital letters. Fixed all letters that were capitalized after the comma.

3-Graphics are needed to drawn with a scientific program, they all need to be drawn in the same format and with an appropriate scientific image, in some of them the name of the  y axis is not given clearly, which is weak for an article image

R/ Good afternoon. Very kind of you for accepting the review of this research. After your comments, we have made all the graphics in origin and now there is a greater uniformity in them. This has allowed us to improve the quality of our research. Thanks for your contributions.

4-As a result, the manuscript is worth publishing in terms of subject and scope, but making these revisions are important in terms of article quality.

R/ Good afternoon. Very kind of you for accepting the review of this research.

Kind Regards

Joaquin.

Reviewer 2 Report

The issues raised in the conclusion section do not include the author's research. The author of the article uses the results of the research.

Author Response

Dear

Thank you for being part of the review of this research.

The issues raised in the conclusion section do not include the author's research. The author of the article uses the results of the research.

R/ Good afternoon. Very kind of you for accepting the review of this research. After your comments, we have corrected the conclusion. Thanks to this, it's now better.

Kind Regards

joaquin

Reviewer 3 Report

This is a valuable piece of scientific work. The employed methods and analytical tools are proper. The used methodology is appropriate. The obtained results can be useful. 

The scientific soundness is limited - as there is a use of typical analytical methods. 

In some figures are present Spanish words - please correct.

Author Response

Dear

Thank you for being part of the review of this research.

This is a valuable piece of scientific work. The employed methods and analytical tools are proper. The used methodology is appropriate. The obtained results can be useful. 

The scientific soundness is limited - as there is a use of typical analytical methods. 

In some figures are present Spanish words - please correct.

R/ Good afternoon. Very kind of you for accepting the review of this research. We have made all the corrections.

Kind Regards

joaquin.

Reviewer 4 Report

The reviewed work concerns the application of pyrolytic analysis (TG-GC/MS) for the evaluation of organic compounds in medical plastic waste generated in the different hospitals of Cartagena-Colombia.
In my opinion, the manuscript was fairly prepared, the results of the study were presented in an understandable way and the results of the research work were thoroughly evaluated. On this basis, I support the authors' request that the manuscript be accepted for further editorial work.

Author Response

Good afternoon.

Very kind of you for accepting the review of this research.

Kind Regards

joaquin

Reviewer 5 Report

Remarks to the authors:

- Page 3, sixth line from the top - footnotes are missing in the text: [41] and [42]

- There is no reference [48] in the text

- Page 18, second line from the top, "...their percentage amounts were low (0.06>)..." should be "... (>0.06).."

- page 20 seventh line from the top - bibliographic footnote [57] appears before [56] - please check the order of the remaining footnotes in the text

- Figure 5 and 6 - the graphical quality (resolution) of the graphs is very low, the lack of a graph grid makes it difficult to read the data, what do the values on the vertical axis mean

- Figure 7 - 12. What does "Date" mean? Please specify units or write [-] if the values are dimensionless. The titles of the tables are duplicated (in the description of the drawing at the bottom and workspaces in the chart)

- General remarks:  "Table 1" (bold) and "table 2" (italics not bold) - please harmonize (also subsequent ones) in accordance with the Template and MDPI guidelines

- General note - headings: "3.3.1. possible reaction..." should be "3.3.1. Possible reaction..."

- General note - chemical formulas in tables, figures and text An example of "...Propane (C3H8)..." (No subscript) should be: "...Propane (C3H8)..."

Author Response

Dear

Thank you for reviewing this research. Each of your contributions was of great value to us.

1- Page 3, sixth line from the top - footnotes are missing in the text: [41] and [42]

R/ Good afternoon. Very kind of you for accepting the review of this research. We have made all the corrections.

2- There is no reference [48] in the text

R/ Good afternoon. Very kind of you for accepting the review of this research. We have made all the corrections.

3- Page 18, second line from the top, "...their percentage amounts were low (0.06>)..." should be "... (>0.06).."

R/ Good afternoon. Very kind of you for accepting the review of this research. We have made all the corrections. The correct form is ≤0.06. Since the concentrations of permanent gases during pyrolysis were less than or equal to 0.06%.

4- page 20 seventh line from the top - bibliographic footnote [57] appears before [56] - please check the order of the remaining footnotes in the text

R/ Good afternoon. Very kind of you for accepting the review of this research. We have made all the corrections.

5- Figure 5 and 6 - the graphical quality (resolution) of the graphs is very low, the lack of a graph grid makes it difficult to read the data, what do the values on the vertical axis mean

R/ Good afternoon. Very kind of you for accepting the review of this research. We have made all the corrections.

6- Figure 7 - 12. What does "Date" mean? Please specify units or write [-] if the values are dimensionless. The titles of the tables are duplicated (in the description of the drawing at the bottom and workspaces in the chart)

R/ Good afternoon. Very kind of you for accepting the review of this research. We have made all the corrections.

7- General remarks:  "Table 1" (bold) and "table 2" (italics not bold) - please harmonize (also subsequent ones) in accordance with the Template and MDPI guidelines

R/ Good afternoon. Very kind of you for accepting the review of this research. We have made all the corrections.

8- General note - headings: "3.3.1. possible reaction..." should be "3.3.1. Possible reaction..."

R/ Good afternoon. Very kind of you for accepting the review of this research. We have made all the corrections.

9- General note - chemical formulas in tables, figures and text An example of "...Propane (C3H8)..." (No subscript) should be: "...Propane (C3H8)..."

R/ Good afternoon. Very kind of you for accepting the review of this research. We have made all the corrections.

Kind Regards

joaquin